# Repetitive training enhances the pattern recognition capability of cultured neural networks

Wen-Wei Shao[1,2,3☯], Qi Shao[1,2,3☯], Hai-Huan Xu[1,2,3☯], Guan-Ji Qiao[1,2,3], Run-Xuan Wang[1,2,3], Zhi-Yun Ma[1,2,3], Wei-Wei Meng[1,2,3], Zhuo-Bin Yang[1,2,3], Yun-Liang Zang[1,2,3*], Xiao-Hong Li [1,2,3*]

1 Academy of Medical Engineering and Translational Medicine, Tianjin University, Tianjin, China, 2 Haihe Laboratory of Brain - Computer Interaction and Human-Machine Integration, Tianjin, China, 3 State Key Laboratory of Advanced Medical Materials and Devices, Tianjin University, Tianjin, China

☯ These authors contributed equally to this work.
* xhli18@tju.edu.cn (XL); yunliangzang@tju.edu.cn (YL)

## Abstract

Cultured neural networks *in vitro* have demonstrated the biocomputing capability to recognize patterns. However, the underlying mechanisms behind information processing and pattern recognition remain less understood. Here, we developed an *in vitro* neural network integrated with microelectrode arrays (MEAs) to explore the network's classification capability and elucidate the mechanisms underlying this classification. After applying different stimulation patterns using MEAs, the network exhibited structural alterations and distinct electrical responses that recognized various stimulation patterns. Alongside the reshaping of network structures, repeated training increased recognition accuracy for each stimulation pattern. Additionally, it was reported for the first time that spontaneous networks after stimulation are more closely related to the structures of evoked networks. This work provides new insights into the structural changes underlying information processing and contributes to our understanding of how cultured neural networks respond to different patterns.

**Data availability statement:** All electro-physiological data files and analysis code data are available from https://github.com/YunDid/Pattern-recognition/tree/master/pattern_recognition.

**Funding:** This work was supported by the National Key Research and Development Program of China (2021YFF1200800 to X.L., 2023YFF1204200 to Y.Z.), the National Natural Science Foundation of China (82171861 to X.L., 62476197 to Y.Z.). The funders had no role in study design, data collection and analysis, decision to publish, or preparation of the manuscript.

**Competing interests:** The authors have declared that no competing interests exist.

## Author summary

Recent advancements in cell-culture methodologies and microelectrode technology have facilitated the development of lab-based neural networks that replicate certain aspects of living brain function. These networks are capable of interacting with external environments through encoding and decoding processes, demonstrating brain-like functions such as memory retention, logical reasoning, and pattern recognition. In addition, they offer an ethical and controllable alternative to studying animal or human brains. Despite their potential for computational tasks, the mechanisms underlying their information processing and learning remain insufficiently understood. In this study, we trained cultured neural networks to classify distinct patterns using microelectrodes. Repeated training sessions resulted in improved classification accuracy and induced structural changes in the network's connections. These findings provide new insights into how biological neural networks adapt through structural modifications, contributing to a deeper understanding of memory formation processes. This research underscores the potential of cultured neural networks as accessible models for investigating brain-inspired computing, neurological disorders, and memory mechanisms, while circumventing ethical concerns associated with animal or human research. By bridging the gap between the functional capabilities and underlying processes of these networks, we aim to foster the development of technologies that emulate biological intelligence and enhance our understanding of brain function.

## 1. Introduction

In recent years, there has been growing interest in the biological computation capabilities of *in vitro* cultured neural networks. These networks are increasingly recognized as valuable platforms for investigating fundamental neural processes. Cultured neural networks exhibit biological computation functions similar to those of the brain functions, including selective adaptation [1,2], parallel memory storage [3,4], logical operation [5], and spatiotemporal pattern identification [6–8]. Additionally, cultured networks can help avoid using humans and animals as experimental subjects and address the technical challenges associated with navigating the intricate architecture of different brain regions.

The study of changes in network connectivity has become a critical tool for investigating memory, information processing, and storage within neural networks. Alterations in the strength and patterns of synaptic connections are thought to underlie memory formation, with synaptic plasticity (such as long-term potentiation and depression) playing a key role in encoding, storing, and recalling information [9–11]. Furthermore, changes in the functional connectivity between neurons are integral to how networks process and store information. Understanding these changes allows researchers to explore how neural networks adapt to new stimuli and consolidate memory [12–14]. This makes the analysis of connectivity an important metric when

investigating the underlying mechanisms of learning and memory in cultured neural networks, as it directly correlates with changes in network behavior and the ability to recognize and process patterns.

Over the past several years, cultured neural networks have attracted increasing attention. Using the reservoir computing framework of brain organoids, Cai et al. conducted spatiotemporal electrical stimulation experiments and verified that unsupervised training enhanced the capabilities of speech recognition and non-linear prediction, which was related to the reshaping of functional network structures [9]. Yang et al. observed that the neuronal firing patterns gradually transitioned towards periodic synchronous bursting after training [12], which plays an important role in neural signal transmission, synaptic plasticity, and network-generated learning [10–13]. Other studies have identified the role of modulating synaptic connections and neuronal firing dynamics in memory consolidation [14–16]. Cultured neural networks are also capable of interacting with external environmental stimuli to precisely control robot movements [17] and autonomously avoid obstacles [18]. By embedding a cultured network into a virtual game, Kagan et al. demonstrated that the network could be trained to play ping-pong within 5 minutes. This research has drawn widespread attention [19].

Despite the progress made in the biological intelligence of cultured neural networks, the biological mechanisms underlying the emergence of the intelligence remain poorly understood. In particular, previous studies did not comprehensively explore the changes in network connections in parallel with the training process and the resulting improvements in "intelligence" [6,7,9,19–23]. In this work, we developed a cultured *in vitro* neural network integrated with microelectrode arrays (MEAs) to test its classification capability and elucidate the underlying mechanisms. We comprehensively analyzed both the evoked and spontaneous functional network structures before, during, and after the training process. We explored the impact of repeated training on the recognition capabilities of the cultured neural networks over three days of training. Additionally, we examined the interplay between evoked and spontaneous functional networks, which plays a critical role in memory consolidation.

## 2. Results

### 2.1. Cultured networks show specific responses to recognize different stimulation patterns

The primary cortical cells from E18 mouse embryos were plated onto the MEAs (Fig 1a), where they grew numerous dendrites and axonal connections over different days *in vitro* (DIV), forming a dense neural network (S1 Fig). The neuronal network began exhibiting spontaneous spikes and synchronous bursts after 14 days of culturing [14,24] (S2 Fig). Specifically, we clarified that only when the neuronal network reaches mature development, characterized by spontaneous activity and the ability to produce evoked responses upon stimulation, are the conditions met for conducting experiments (Fig 1b). This criterion ensures the reliability and relevance of the experimental data. We then selected the stimulated electrodes based on the post-stimulus time histograms (PSTH) of neuronal responses (see Methods), applied two different training stimulation patterns, and tested the pattern recognition capability of the cultured network (Figs 1c and S3). We subsequently used the logistic regression method to analyze the evoked responses of the network and evaluated its recognition capability (Fig 1d and 1e). Our results show that the two stimulation patterns can be well classified based on the network's response properties after training. Likely due to the high accuracy achieved on the first day, further training did not significantly improve classification performance.

Next, we investigated whether stimulation-pattern-specific neurons existed. We observed three groups of neurons in the cultured network. As shown in Fig 1f, some neurons exhibited stronger responses to one particular stimulation compared to the other. Other neurons showed nearly identical responses to both types of stimulation. There was also a group of neurons that didn't respond to either stimulation. The PSTHs of neuronal responses are shown in Fig 1g. Neurons exhibiting preferred responses to either L or X demonstrated a noticeably elevated frequency of response compared to the other stimulation pattern, while non-specific neurons exhibited minimal variance in response to each stimulus. These results support the presence of neurons in the cultured network that can accurately identify and react to specific stimulation patterns, highlighting the network's capability of processing and classifying input information.

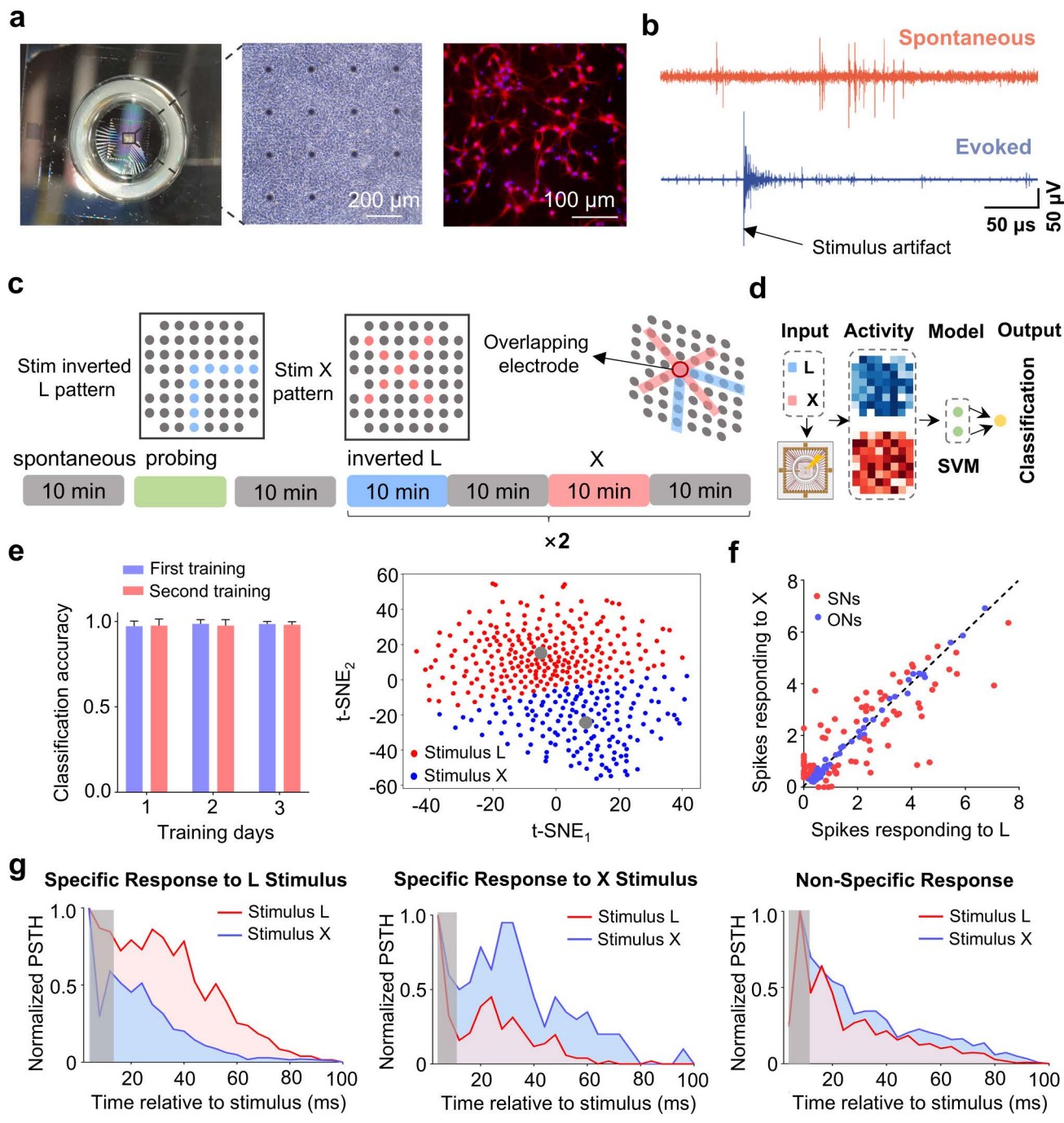

**Fig 1. The response properties of the cultured network. (a)** Cortical neurons were plated on MEAs at DIV 21 on MEAs. Immunofluorescence patterns of the neuronal network using MAP-2: neurons (red), nuclei (blue). **(b)** Spontaneous (top) and evoked response activities. **(c)** Stimulation (training) protocol. Spontaneous activities for 10 minutes (gray period), specific stimulation electrodes for the patterns are selected (green period), training interval L pattern (blue period), training the X pattern (pink period). **(d)** The schematic of pattern classification based on the network responses. SVM, Support Vector Machine. **(e)** Classification accuracy after training with two stimulation patterns (mean±s.m.e., n = 10) and t-SNE plot of neuronal responses. **(f)** The distribution of neuronal responses. The x-axis indicates the number of spikes in response to the interval L stimulation pattern, while the y-axis indicates the number of spikes in response to the interval X stimulation pattern. Red represents stimulation specific nodes (SNs); blue represents the other non-specific nodes (ONs). **(g)** The PSTHs of specific and non-specific responses to stimulation patterns after 100 ms (bin = 2 ms). The grey shaded region represents post-stimulation artifacts.

Our results demonstrated that the two stimulation patterns could be effectively classified based on the response properties of the cultured network after training. To establish a framework for validating the pattern classification capability of cultured neural networks, we initially selected two stimulation patterns. This approach allowed us to focus on investigating the fundamental mechanisms of structural and functional reorganization during training. Recognizing the need for a more comprehensive evaluation of the network's classification capabilities, we subsequently increased the number of stimulation patterns to explore the processing capacity and mechanisms of the biological neural network in response to more complex inputs.

## 2.2. Changes of neuronal dynamics and functional connectivity of the cultured network after training

We have demonstrated that the evoked responses of the cultured network can classify different input patterns after repeated training. To further elucidate the influence of training on neuronal dynamics, we compared the spontaneous activity of the network before and after training. As shown by the raster plots and population firing rates, the network exhibited more frequent bursting after training (Fig 2a). Statistically, the average firing rate and the burst rate significantly increased, while the inter-spike interval within bursts decreased (Fig 2b). In the untrained control group, the results of spontaneous electrical activity analysis over the same time period showed no significant changes in spontaneous electrical activity (S4 Fig). Meanwhile, neurons in the network became more synchronized after training (Fig 2c). Given the well-established evidence that bursting patterns correlate with changes in synaptic connections for handling and storing information [14,25], we deduce that the network's capability for pattern classification is due to the more frequent and synchronized bursting activities.

We calculated the cross-correlation coefficients between neuronal spontaneous activities as a measure of neuronal connection strengths to construct a functional connectivity network [11,26]. We next analyzed the changes in network connectivity after exposure to different stimulation patterns. Our results show that the network structure underwent reconfiguration after training. Individual neuronal connections could become strengthened, weakened, newly formed, or pruned (Fig 3a). For the network that consistently experienced the same stimulation patterns, the ratio of strengthened neuronal connections was higher than that of the network exposed to different stimulation patterns. In contrast, the ratios of weakened, newly formed, and pruned connections were lower in the networks that experienced the same stimulation patterns. We also calculated the Euclidean distances of the functional networks after training. Compared to the networks exposed to different stimulation patterns, those networks that experienced the same patterns had smaller distances, suggesting that their neurons were "closer" to each other (Fig 3b). These results demonstrate that network connections underwent changes after training, which could underlie the network's capability for pattern classification.

## 2.3. Improved recognition performance with continual learning

The previous results support the capability of the cultured network for binary pattern classification after training. We further challenged the cultured network by designing a paradigm composed of six different stimulation patterns (Fig 4a and 4b). Fig 4c shows the evoked responses of the cultured network following these stimulation patterns. As done previously in Fig 1d, we trained a logistic regression classifier to classify the six different stimulus patterns based on these post-stimulus responses. Fig 4d illustrates the changes in the network's classification accuracy during the three-day training period. After one day of training, the classification accuracy reached 93% for the two-pattern stimulation and classification task. However, when the number of stimulation patterns increased to six, the classification accuracy decreased by approximately 22%. The tradeoff in recognition accuracy is influenced by the number of stimulation patterns, as an increase in the number of classes introduces greater complexity to the classification task. Regardless of the number of stimulation patterns, the classification accuracy continued to improve with repeated training. The

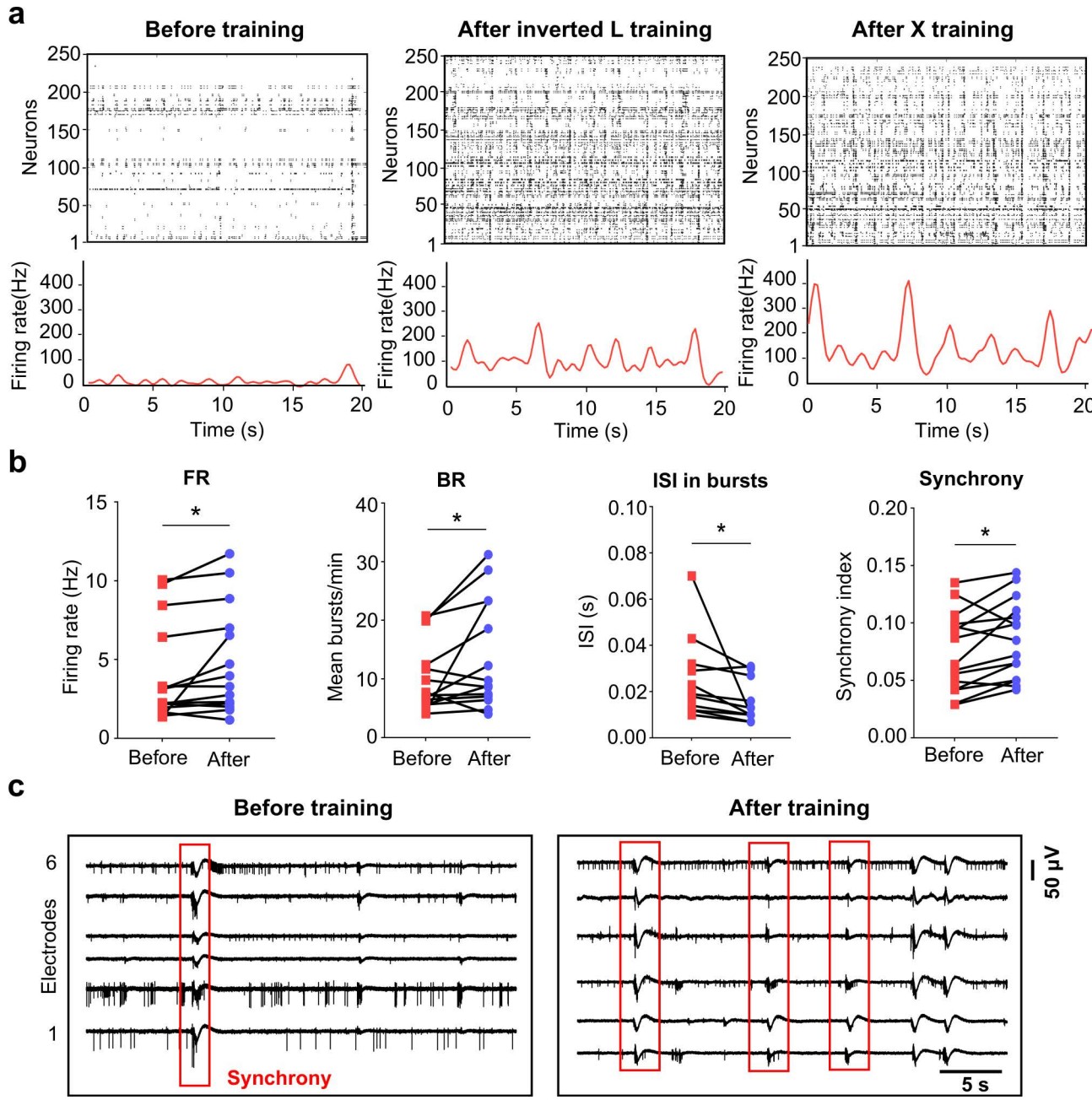

**Fig 2. Training changes spontaneous dynamics of the network. (a)** Raster plots of the network spontaneous activities before and after training. Each vertical bar represents a spike (top). The bottom plot shows the network population firing rate. **(b)** Quantitative indicators of neuronal activity before and after training (firing rate, FR; bursting rate, BR; Inter spike interval within bursts, ISI in bursts; synchrony index. $n = 5$, paired t-test, *$p < 0.05$, **$p < 0.01$, ***$p < 0.001$) **(c)** Synchronous bursts before and after training, where red box represents a synchronized burst.

accuracy increased to 98.2% for the two-pattern classification and to 82.5% for the six-pattern classification. We also calculated the Bayesian information criterion (BIC) to measure the trade-off between model fit and complexity. As indicated by the number of classes that achieved the lowest BIC value, training gradually improved the recognition capability of the network (Fig 4e).

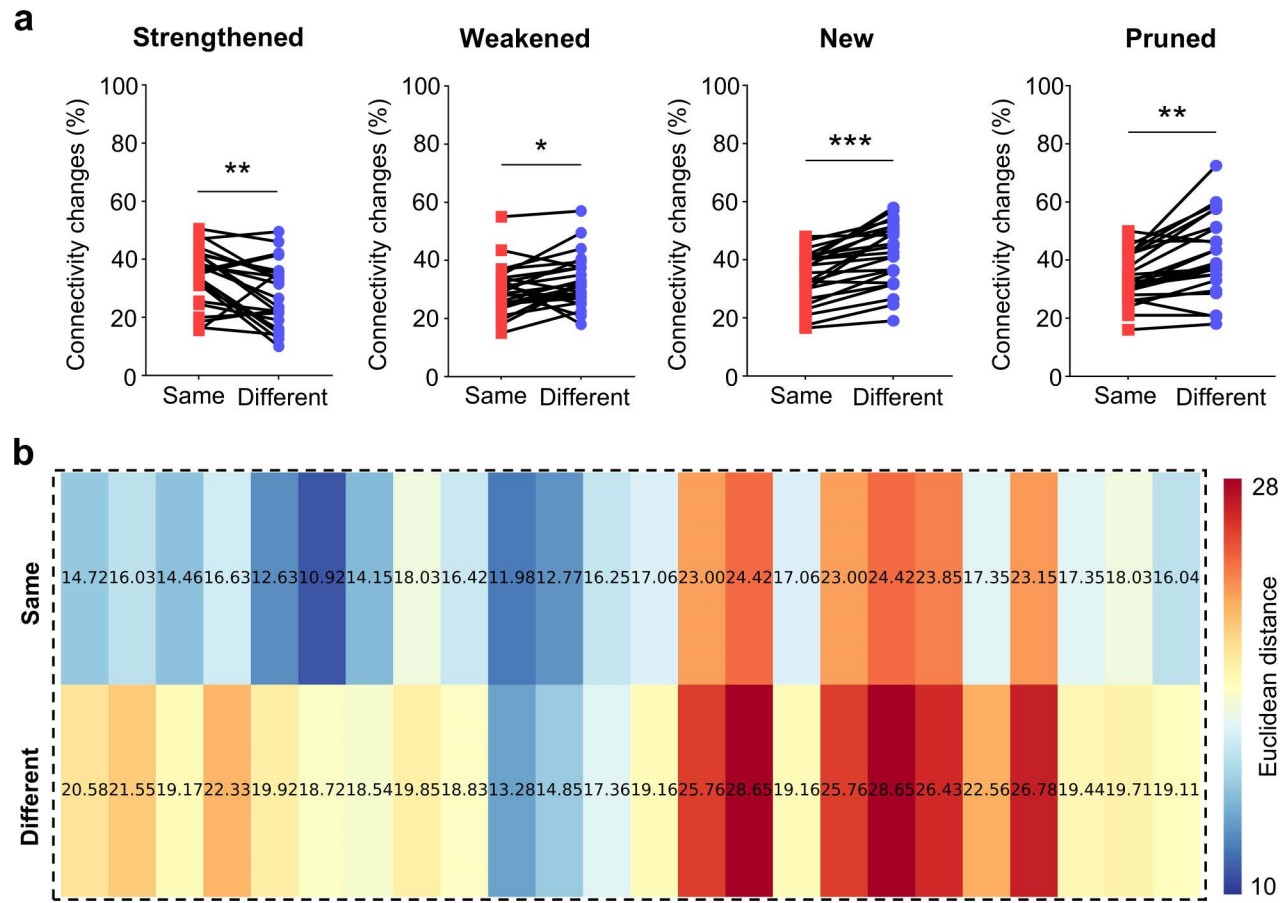

**Fig 3. Functional connectivity changes after experiencing different stimulation patterns. (a)** Ratio of network connectivity changes (strengthened, weakened, newly formed, pruned) between the same and different stimulation patterns. (Same: interval L or X in the first and second rounds of training; Different: interval L vs. X in either the first or second round of training), (mean±s.e.m., n=8, paired t-test, **p<0.01, ***p<0.001). **(b)** Heat maps of Euclidean distance for the networks that experienced the same (above) and different (bottom) stimulation patterns (n=5, ***p<0.001).

## 2.4. Training changed the frequencies of the neural network activity

Similar to the two-pattern training (Fig 2), we observed more frequent and synchronized spontaneous bursting activities in the network after six-pattern training (S5 Fig). To further characterize how repeated stimulation changed the spontaneous dynamics and recognition capability of the cultured networks, we plotted the power spectral density and power spectrogram of each network. Fig 5a displays the power spectrograms of each network's spontaneous activity after different patterns of stimuli. As indicated by the brightening of the δ- and θ-frequency bands, the post-stimulus network showed increased low-frequency neural activities compared to before stimulation (Fig 5b). The phase coupling between spikes and the local field potential in various frequency bands was studied before (Fig 5c) and after training (Fig 5d). We used the Rayleigh criterion to determine the significance of the non-uniform distribution of spikes in circular phase space. Spikes exhibited a strong preference for the δ and θ phases during the narrow phase window centered on their average angle. A p-value of <0.05 indicates a non-uniform distribution, suggesting that spikes and the δ- and θ-frequency signals are strongly coupled [23], indicating that neurons possess the ability to retain stimulation information.

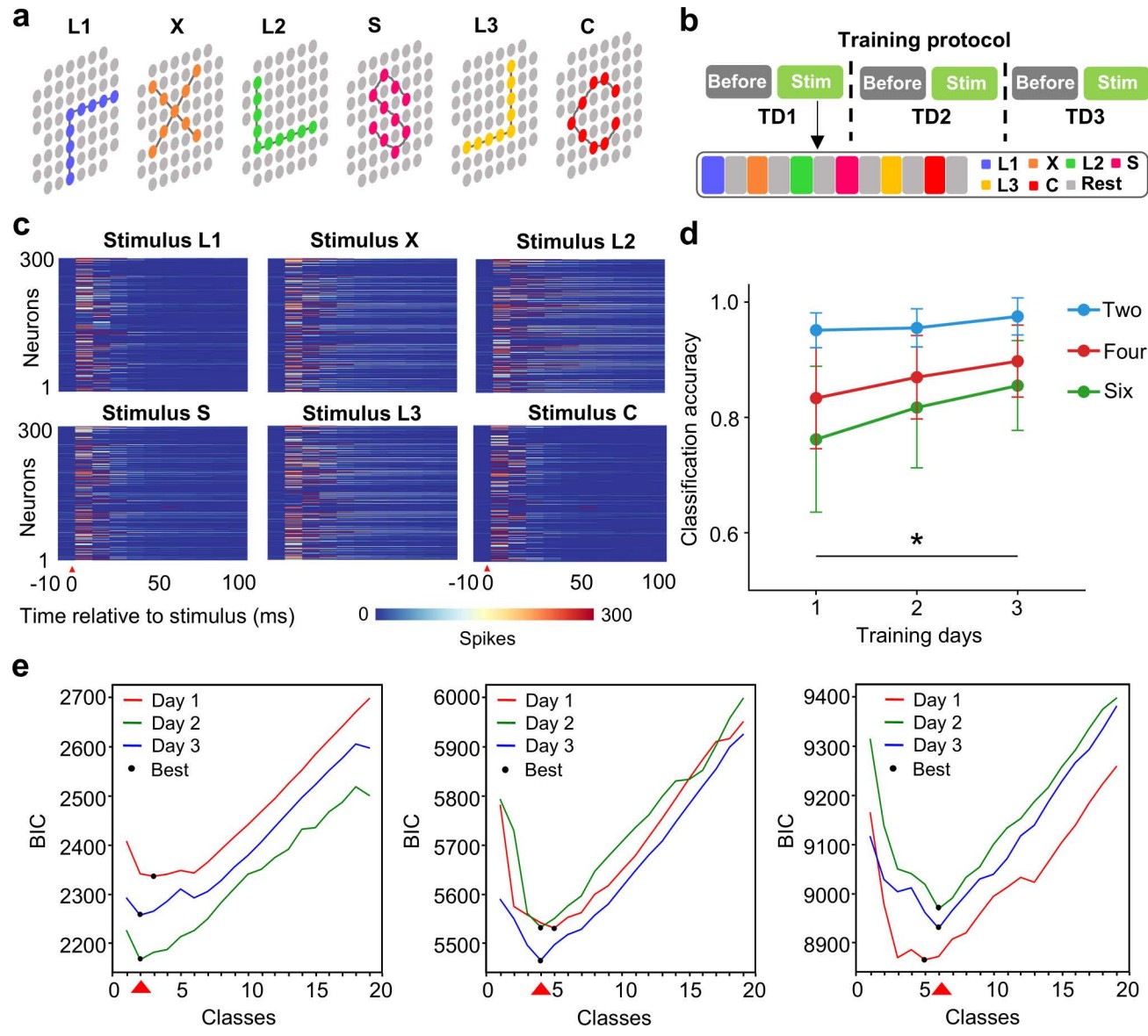

**Fig 4. Improved classification performance by repeated training. (a)** The schematic of the six stimulation patterns. Colored dots represent stimulated electrodes, while the gray dots represent the non-stimulated electrodes. **(b)** The schematic of the training protocol. **(c)** Post-stimulus response activities. The red triangle indicates the timing of the stimulation. **(d)** Increased classification accuracy with training days (mean±s.m.e., n=10, *p<0.05). **(e)** BIC computed for different number of classes. Black dots indicate the optimal number of classes after each-day training, and the red triangle indicates the target number of classes.

## 2.5. Continual changes of network structures during repeated training

During the training process, repeated stimulation can alter synaptic connectivity to store information [17]. Therefore, we conducted an analysis of the alterations in the evoked neural network structure. The results revealed that the properties of the evoked functional networks across the six patterns, including metrics such as the average degree and density, did not exhibit significant variations with respect to the number of training days (Fig 6a). This suggests that network metrics, such as density, do not account for the mechanism underlying the observed improvement in classification accuracy over

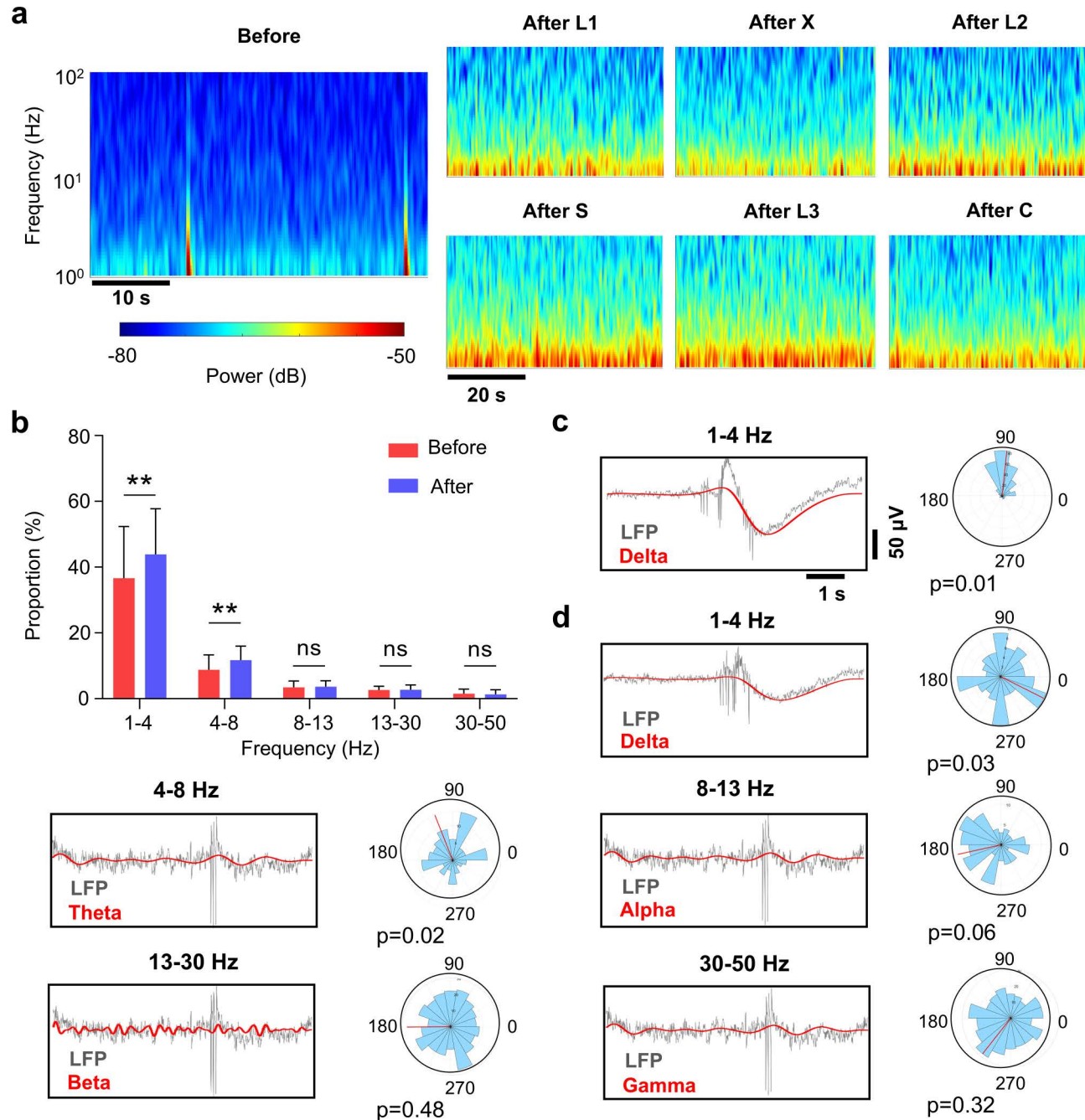

**Fig 5. The impact of training on the frequency of neural network activity. (a)** Time-frequency plot before and after different patterns of stimuli. **(b)** Power spectral density of each frequency band before and after training (Delta: 1-4 Hz; Theta: 4-8 Hz; Alpha: 8-13 Hz; Beta: 13-30 Hz; Gamma: 30-50 Hz), (mean±s.m.e., n=10, *p<0.05). **(c)** The alignment between LFP (gray) and a delta band-pass filter (red) before training. **(d)** The alignment between LFP (gray) and band-pass filters (red) after training. In c and d, the Rayleigh criterion for non-uniformity is used to determine if the peaks are unevenly distributed over the corresponding frequency band period (0°, 360°).

the course of training. To further elucidate this mechanism, we subsequently computed and compared the Euclidean distances between networks subjected to different stimulation patterns during the training process (Fig 6b). We observed that the Euclidean distances between different stimulation patterns increased with training (Fig 6c), potentially supporting

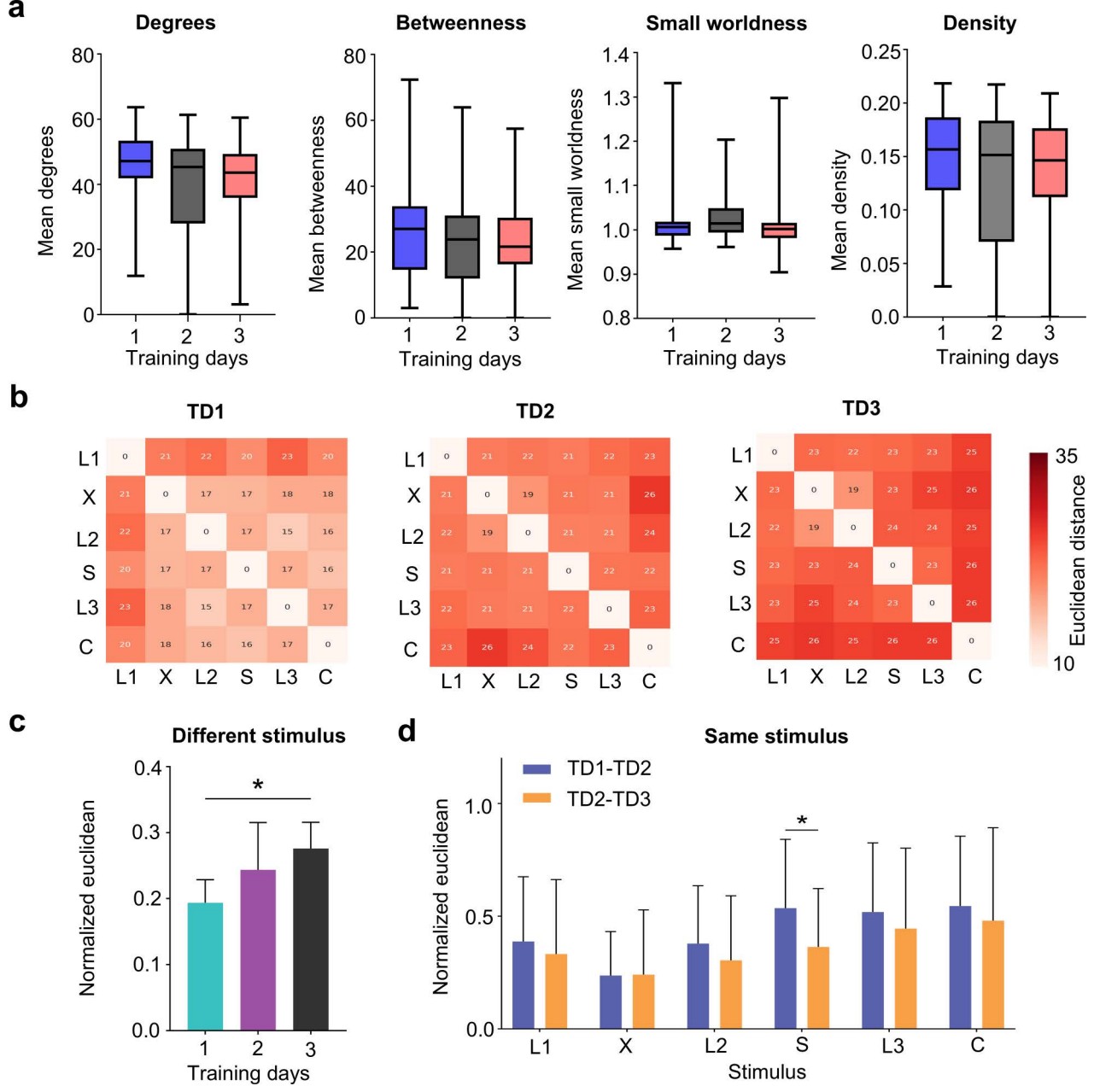

**Fig 6. The network structure changes during repeated training. (a)** Network metric changes with continual learning, including degree, betweenness, small-world-ness, and density (mean±s.m.e., n=10). **(b)** Euclidean distance coefficient matrix between different stimulation patterns after three days of training (Training Day, TD). **(c)** Statistics of normalized Euclidean distance after three days of training different patterns. **(d)** Euclidean distance of the networks that experienced the same type of simulation patterns over two consecutive days (mean±s.m.e., n=10, *$p<0.05$).

the gradual improvement in classification accuracy among different stimulation patterns. In contrast, for the network that experienced the same type of stimulation patterns, the Euclidean distances decreased with continued training (Fig 6d).

We next analyzed how training changed the functional network connections right before and after each stimulation. Fig 7a shows the functional matrix before (spontaneous), during (evoked), and after (spontaneous) each stimulation

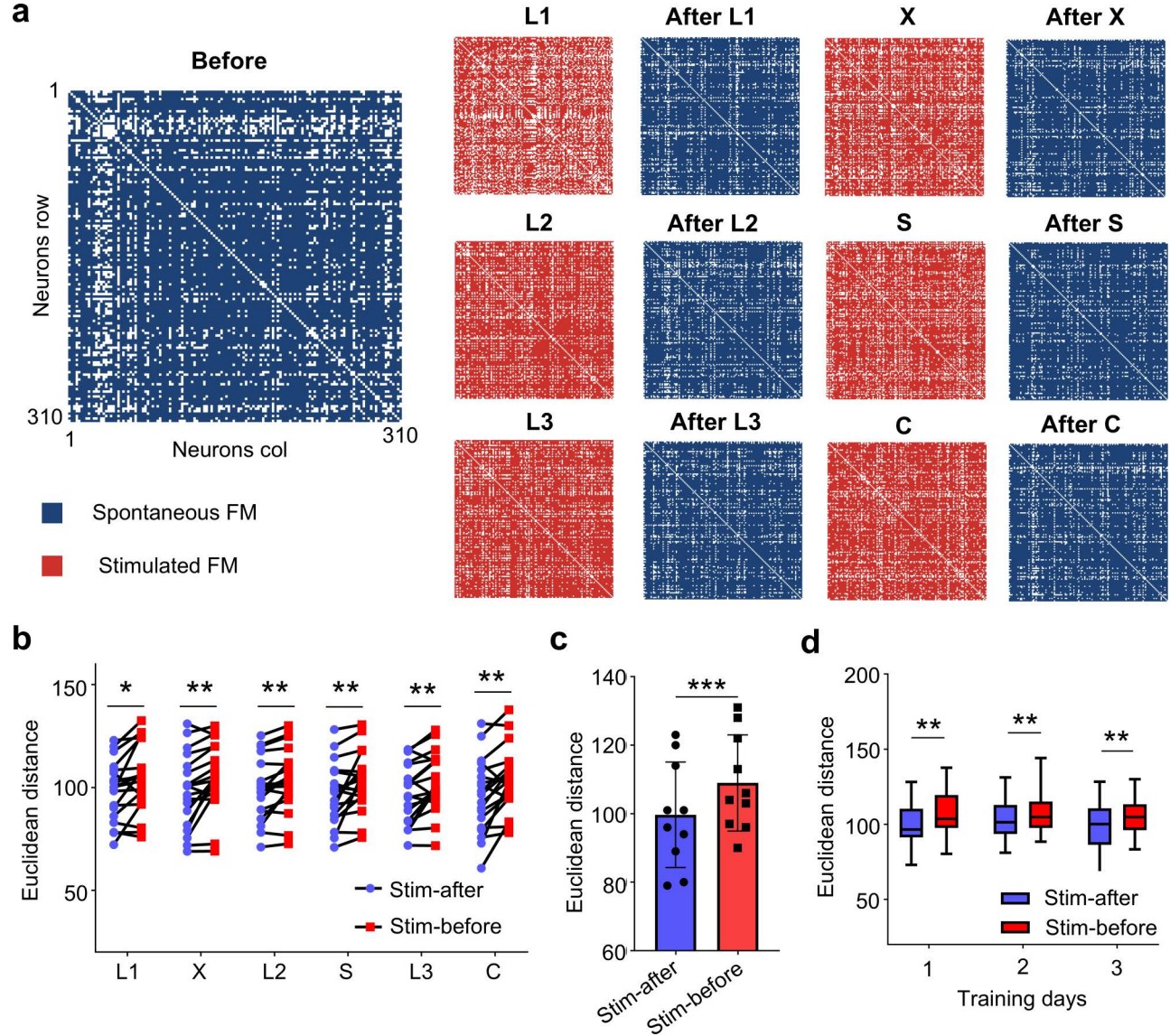

**Fig 7. Comparison of the connectivity between spontaneous- and evoked-functional network. (a)** Spontaneous (blue) and evoked (red) functional matrices before and after training for each stimulus pattern. **(b)** Comparison of the Euclidean distances between the evoked functional networks from each stimulus pattern and the spontaneous ones before and after the stimulation (n = 10, paired t-test, *$p < 0.05$, **$p < 0.01$). **(c)** Statistical analysis of Euclidean distances between functional matrices evoked by all stimulation patterns (n = 10, paired t-test, ***$p < 0.001$). **(d)** Statistical analysis of Euclidean distances between stimulation and post-stimulus (stim - after) as well as between training and pre-stimulus (stim - before) for three days of training (n = 10, paired t-test, **$p < 0.01$).

pattern. We calculated the Euclidean distances for the functional networks during training compared to the networks of pre-stimulus spontaneous activity and post-stimulus spontaneous activity (Fig 7b-d). Our results suggest that the Euclidean distance of the networks between training and post-stimulus is significantly smaller than the distance between training and pre-stimulus. This trend persisted throughout the training process, indicating that the network after training is closer in structure to the network during training.

## 3. Discussion

In this study, we have shown that repeated training can improve the pattern recognition capability of cultured neural networks and establish a link between gradually improved recognition accuracy and changes in network structure. To our knowledge, for the first time, we examined the association between the evoked and spontaneous network structures, revealing a closer alignment between the evoked and post-stimulus spontaneous structures.

Previous studies have shown that adaptive behaviors can emerge spontaneously from embodied cultured neurons [27]. Through closed-loop integration with incentive-punishment rules, better task-completion capabilities can be achieved by adjusting network structures and correcting erroneous behaviors [19]. Unsupervised training can also enhance the ability of cultured brain organoids to recognize speech, with classification accuracy reaching around 80% [9]. Although these studies demonstrated improved classification accuracy through training, the contributions of network structure were not well characterized. In our study, the evoked response of the global network were collected and characterized that each stimulus pattern could induce different network response patterns by using a logistic regression model to classify the response features evoked by stimulus.

The stimulus-response relationship holds considerable significance in facilitating the execution of certain tasks within biological networks [28,29]. Utilizing optical and electrical stimulation coding patterns, previous studies demonstrate that cultrured neural netowrks have spatially specific responses and familiarity detection capabilities [8,30]. While optical stimulation offers precision, they pose safety risks due to viral transfection. In this study, the candidate stimulation electrodes were used to create partially overlapping spatial stimulation patterns, characterized by closer spatial proximity, overlapping stimulus points, and increased complexity while preserving visual characteristics. This approach yielded accuracy levels comparable to optical stimulation.

It is worth noting that the cultured neural network can initially recognize two types of stimulation patterns with an accuracy of over 90%. As we fed more stimulation patterns into the cultured network, the classification accuracy decreased (Fig 4d), but the recognition performance gradually improved with training. A similar phenomenon has been observed when training convolutional neural networks to recognize complex images [31,32]. Likewise, the human visual system also experiences a decrease in recognition capabilities as the complexity and similarity of the input information increase [33,34]. These results suggest that biological brains may not be as powerful as we previously thought. Instead, similar to AI techniques, the brain may also face recognition issues when dealing with highly complex visual information.

The network structure modifies and reshapes itself to adapt to the input stimulation. While we have used the cross-correlation (CC) method to assess functional connectivity, future work will explore the use of Maximum Entropy (MaxEnt) models to estimate functional connectivity during stimulation and infer both excitatory and inhibitory connections [35]. Within a network, neurons communicate through synapses and adjust the connection weights to process information, a phenomenon referred to as synaptic plasticity. Similar to previous findings [36,37], the cultured networks, based on the measured electrical signals, gradually adapt to external stimulus information and achieve a new equilibrium in the form of elevated firing rates, burst rates, and network synchronization (Fig 2). Increased burst activities have been found to promote information processing and transmission in the neural network [36,37], which may underlie the improved recognition capability of our cultured network. Changes in electrical activity reflect changes of network connections. However, in experiments, we still lack a method to measure synaptic connections directly. In this context, building a functional neural network provides an alternative approach to explore changes in network connections after training. We computed the similarity of the evoked network structure during the training process. The Euclidean distance of networks with the same stimulus pattern gradually decreases, while it gradually increases among different stimulation patterns. These results support the theory that the network possesses parallel information processing capabilities and that different information processing structures do not interfere with each other [3,38].

In addition, our findings indicate a closer alignment between the evoked network structure and the post-stimulus spontaneous network structure, arguing that memory exists in the spontaneous network structure after training. Although it has been proven that evoked and spontaneous neural networks have similar network burst activity patterns [39], our

study explores the mechanism of intelligence enhancement through training. The presence of diverse stimulation patterns results in heightened neural network response activity, yet uncertainty persists regarding the gradual establishment of stable and unique information-processing pathways and response patterns for each stimulus pattern. The definitive identification of the inherent information processing pathways for each stimulus pattern and network structures akin to the spontaneous mode remains elusive, which encounters challenges due to the network's highly dynamic state during the training process [40]. Future investigations may require a more profound engagement with graph theory and information theory to elucidate cultured neural networks' diverse information processing modalities and devise strategies for developing and consolidating network structures.

Lamberti et al. employed mutual information (MI) to quantitatively assess memory and prediction efficiency in neural networks, providing a detailed and mathematically rigorous framework for analyzing network dynamics [41]. While their method is highly effective in measuring prediction accuracy, our study places greater emphasis on the relationship between changes in functional connectivity within the network and the subsequent improvements in recognition performance. In contrast to Joost et al., we did not explore the application of MI for evaluating memory. Nonetheless, their approach offers valuable insights into memory dynamics, and we plan to incorporate MI-based methods in future work to further investigate both memory and prediction aspects of our trained networks [3].

Our work was performed in 2D cultured neural networks. The inherent self-organizational capacity of stem cell progeny affords them the ability to differentiate into 3D organoids, capable of mimicking various human organs or tissues [42]. Unlike conventional 2D cell differentiation or primary cultures, brain organoids offer a more faithful representation of *in vivo* conditions, demonstrating superior fidelity in terms of cellular diversity, the intricacies of 3D physiological environments facilitating cell development, the orchestration of complex neural network dynamics, and increased neuronal maturation under low-intensity ultrasound [43,44]. It is reasonable to anticipate that their computational efficacy will surpass that of conventional 2D cultured neural networks utilized in prior investigations. However, using 2D cultured networks should not affect our main findings in this work, since the neural mechanisms underlying the improved recognition capability should remain valid.

In summary, by repeatedly training the cultured neural networks to recognize different stimulation patterns, our work suggests that the network structures converge when exposed to identical stimulation patterns but diverge when encountering different stimuli. Additionally, we discovered a closer alignment between evoked and spontaneous network structures following stimulation. Our results not only advance the understanding of information processing in the brain but also offer promising avenues for the exploration of intelligent behaviors using cultured neural networks.

## 4. Materials and methods

### 4.1. Ethics Statement

The animal study protocol was approved by the Ethics Committee of Tianjin University, with approval number: TJUE-2023–221.

### 4.2. Culturing primary cortical neuron

For this experiment, 4–5 fetal rats were obtained from 18-day gestation Viton Lever (Wistar) rats and placed in D-Hank's balanced salt solution (D-HBSS). They were then transferred to an ultraclean table as soon as possible. The next step involved isolating the cortical tissue. Using ophthalmic scissors and forceps, the skin attached to the skull was carefully removed. The fetal rat skull was meticulously opened, and the fetal rat scalp was lifted to avoid damaging the brain tissue. The intact brain was then separated along its ventral surface using forceps. The left and right cortices were peeled off and placed in D-HBSS to fully shear the tissue. All procedures were performed on ice to maintain tissue viability. Subsequently, the cortical neurons were dissociated. The tissue was digested into single cells by adding 0.125% trypsin and incubating at 37°C for enzymatic digestion, with shaking every 4 minutes. After 10 min, the digestion was terminated by

adding 4 ml of growth culture medium (2% B27 + 1% PS + 0.5% Glutmax). The cell suspension was filtered using a 100 μm cell strainer, and the cell concentration was adjusted to 1000–1500 cells/μL. For cell growth, 80–90 μL of cell suspension was inoculated onto double-coated multi-electrode arrays (first layer: poly-D-lysine 100 μL/mL, second layer: laminin 50 μL/mL) and placed in an incubator at 37°C and 5% $CO_2$ for 4 hours to allow the cells attachment. After 4 hours, the neural maintenance medium was changed, and thereafter, half of the medium volume was replaced every 3 days until the experiment's conclusion [45].

### 4.3. Data acquisition with MEAs

Neuronal electrophysiological signals were acquired using the MEA2100-Mini-Systems (Mini, Multi-Channel Systems-MCS, Reutlingen, Germany). The microelectrode array comprises 59 TiN/SiN planar circular electrodes (30μm diameter and 200μm electrode center spacing) arranged in an 8 × 8 electrode array in a square grid, with one electrode serving as the reference. The MCS device can be placed in an incubator, enabling long-term experiments while maintaining the required temperature and $CO_2$ environmental conditions for neurons. Electrophysiological signals were acquired at a sampling rate of 25 kHz via the commercial software Mini-Multichannel Experiment, which also facilitated online visualization and raw data storage.

### 4.4. Spike Detection and Spike Sorting

To extract single-unit activity from the raw MEA recordings, spike sorting was performed. Initially, the raw signals were band-pass filtered between 300 ~ 3000 Hz using the Detect Spikes function in NeuroExplorer (NEX) to remove background noise and improve the signal-to-noise ratio. A threshold of 5 times the standard deviation was then applied to identify spikes in the filtered data. For spike sorting, we used the Sort Spike function in NEX based on the SpyKING CIRCUS algorithm [46–48], which classifies the detected spikes according to their waveform characteristics. This method allows for effective separation of spike events originating from different neurons. The final output consists of spike time data corresponding to individual neurons.

### 4.5. Burst and network burst detection

Bursts are important and high frequency neuronal firing patterns that generally appear after 1 week of *in vitro* culture of neurons and mark the maturation of neuronal network development [49]. In this experiment, a maximum interval detection method was used to identify network bursts, which is embedded in the NEX and contains five parameters: maximum interval between first fronts, minimum number of fronts, duration, outburst interval and maximum internal interval between fronts [45,50]. Network bursts are identified based on individual channel bursts and are defined as simultaneous network bursts when at least 20% of the electrodes are active within a specific time window [39].

### 4.6. Experimental protocol

The experiments were all started after 21 days culture of *in vitro* neuronal network. A 10-minute recording of spontaneous neuronal activity was conducted prior to the experiments. This preliminary recording ensured that both neurons and the apparatus were stabilized, thereby minimizing noise that could affect the experimental outcomes. Pre-experimentation involved bidirectional voltage stimulation (500 mV, 1 Hz), as determined from preliminary tests (S6 and S7 Figs). Random sequential stimulation of all electrodes was performed. The PSTH of the remaining electrodes was calculated by stimulating each electrode, assessing the stimulation response. Candidate stimulation electrodes were identified based on PSTH, thereby avoiding discrepancies in stimulus response evoked by the two selected images. Following random stimulation, a 10-minute recording of spontaneous neuronal activity was conducted to allow neuronal firing to stabilize. Prior to initiating the training experiment, 10 random stimulations of the

selected patterns were performed. Training involved the two selected spatial stimulation patterns, with 40 stimulations per round for each pattern, 10 rounds of training at 1 Hz (biphasic square voltage wave, amplitude ±500 mV, duration 200 μs), followed by a 10-minute rest. The training for each pattern was conducted in two alternating cycles, with two repetitions for each pattern.

### 4.7. Specific nodes (SNs) detection

To investigate the mechanisms by which cortical neuron networks process and store complex information, this study proposed a computational method to identify neuron nodes with specific responses to input information. The two-pattern PSTH value for each neuron pair were calculated, and the PSTH of each neuron for each stimulus type was normalized using the maximum value method. This normalization facilitated uniform comparison between the two types of stimuli. A two-dimensional scatter plot was created, representing the normalized response strength of each neuron to the two patterns. The distance from each node to the diagonal line (which indicates equal response strength to both patterns) was calculated. Nodes with a diagonal distance greater than the average distance of all nodes to the diagonal were identified as specific response nodes. This type of node has specific response strength to specific input information, which is consistent with the grandmother cell hypothesis.

### 4.8. Functional network construction and evaluation

In order to investigate the effects of stimuli on the neuronal network connectivity and to assess the dynamic changes within the network, this study employed a widely used Cross-correlation algorithm to construct a functional network topology matrix [26,51–53]. The correlation between the spike trains of two neurons based on MEAs acquisition was calculated by the time-delay method (time window as 50ms, bin as 2ms). One neuron needs to be used as the target electrode and the other as the reference electrode, and the two time series are aligned by start time. A 2 ms long time window (bin) is created centered on the time when each spike in the reference electrode appears, which is used as the reference to detect whether the spikes in the target electrode are included in the window, and the value in each window is the number of spikes in the target electrode falling into the window. Cycle through each spike point in the reference electrode to obtain the corresponding numerical sequence. Finally, the normalized result is expressed as the number of interrelationships between the target electrode and the reference electrode. Here is the cross-correlation function for evaluating the spike train for each pair of electrodes (x, y) [10]:

$$C_{xy}(\tau) = \frac{1}{N_x N_y} \sum_{s=1}^{N_x} \sum_{t_i=\tau-(\frac{\Delta\tau}{2})}^{\tau+(\frac{\Delta\tau}{2})} x(t_s)y(t_s + t_i) \qquad 1$$

where $t_s$ is the duration of each spike in train x, $N_x$ is spike's total number in x and $N_y$ represents the spike's total number in y.

The resulting correlations were then normalized to determine the strength of the functional synaptic connections between the two neurons and the causal transmissibility. A correlation coefficient close to 0 indicated either no synaptic connection relationship or weak connection, whereas a coefficient close to 1 signified a stronger synaptic connection. To address the presence of weak or spurious connections in the functional networks, a threshold method with mean + 1 SD deviation is used to preserve the real connection relationship between neurons. Despite this, some weakly connected edges were still removed. To enhance accuracy, the mean and variance method was subsequently employed to re-evaluate and screen these sub-threshold connections (S8 Fig), thereby improving the accuracy of the functional network [54]. According to the constructed functional topology matrix, four indicators, network degree, density, small-worldness and betweenness, were calculated separately. A small-worldness value close to or greater than 1 indicated that the network exhibited efficient parallel information processing capabilities characteristic of a small-world network [55].

## 4.9. Euclidean distance (ED)

Euclidean distance is one of the most common methods to calculate the distance, in order to evaluate the similarity of two functional networks, based on the constructed functional neuronal network matrix, the distance of the two matrices corresponding to the weight coefficient is calculated [37,38,3], if the distance is larger, the structure of the two functional neural networks is different, if the distance is smaller, it indicates that the structure of the two functional neural networks is more similar. The ED between connectivity matrices at time $t$ and time $t_0$ can be expressed as

$$ED(t) = \sqrt{\sum_{i=1}^{n} \sum_{j=1}^{a} [S_{ij}(t) - S_{ij}(t_0)]^2} \tag{2}$$

where the $S_{ij}$ is represented as the weight coefficient in column j of row i.

## Supporting information

**S1 Fig. Neuronal network structures during development.** (a) Structural changes of neurons cultured in vitro at different developmental time points. (b) Functional network connectivity, where nodes represent neurons, node size represents node degree, and the larger the node degree, the color is close to red, and the lower is close to blue.
(TIF)

**S2 Fig. Spontaneous activity during neuronal development.** (a) Original waveform graph of spontaneous electrical activity in neural networks at different developmental time points. (b) Raster plot of spontaneous activity in neural networks, where the black lines represent a single spike, and the red lines represent the firing rate of the neuronal network. (c) Quantitative indicators of activity in neural networks cultured in vitro at different developmental days (Number of spikes; number of bursts; interburst interval, IBI; spikes in a burst) (mean±s.m.e., n = 5, *p < 0.05, **p < 0.01, ***p < 0.001).
(TIF)

**S3 Fig. Probing detailed experimental steps and specific experimental parameters for training.** (a) Select stimulation points and combine them into a stimulation pattern. (b) PSTH (Peri-Stimulus Time Histogram) response of neural networks induced by different stimulation. The higher the induced response intensity, the larger the PSTH area. (c) In the corresponding response activity diagram after stimulation in (b), the stronger the response, the closer to red; the weaker the response, the closer to blue. (d) Specific training stimulation paradigm.
(TIF)

**S4 Fig. The spontaneous dynamics of the network without training.** A comparative analysis was conducted about the spontaneous neural activity of the untrained control group. The control group was subjected to electrophysiological signal acquisition at intervals identical to those of the trained group, albeit in the absence of any stimulus-based training regimen. During data analysis, the control group was maintained under equivalent experimental conditions, encompassing an identical neurodevelopmental temporal framework.
(TIF)

**S5 Fig. Spontaneous network dynamics under training with multiple stimulation patterns stimulation.** (a) Raster plot of spontaneous activity in neural networks, where the black lines represent a single spike, and the red lines represent the firing rate of the neuronal network. (b) Quantitative indicators of neuronal activity before and after training (mean firing rate, MFR; mean bursting rate, MBR; interburst interval, IBI; interspike interval (ISI) in a burst; synchrony index) (n = 10, paired t-test, *p < 0.05, **p < 0.01, ***p < 0.001).
(TIF)

**S6 Fig. Amplitude gradient testing of neuronal network activity.** (a) Neuronal network activity is induced in ascending order of amplitude; the darker the color, the stronger the evoked firing rate. The lower right corner displays the average

discharge rate of the neuronal network under different amplitude stimulations (b) In contrast to the stimulus amplitude order in (a). (c) The induced firing rate of the neuronal network under different numbers of stimulus and different amplitude stimulations (1Hz). (mean±m.s.e., n = 5).
(TIF)

**S7 Fig. Frequency changes the dynamics of the neural network.** (a) Different stimulation frequencies induce activity in the neural network. (b) The evoked response of neural networks under different frequency stimulation was compared. (c) Different stimulus frequencies correspond to the evoked response under different stimulus.
(TIF)

**S8 Fig. The process of constructing a functional network.** (a) The cross-correlation algorithm is used to calculate the correlation between the spike train of two neurons as the edge weight. (b) Double Threshold (DDT) algorithm operating principle applied to a simple network.
(TIF)

## Acknowledgments

The authors would like to thank the Prof. Jianguo Zhang (Southern University of Science and Technology) for his valuable comments on improving the paper.

## Author contributions

**Conceptualization:** Yun-Liang Zang, Xiaohong Li.

**Funding acquisition:** Yun-Liang Zang, Xiaohong Li.

**Investigation:** Wen-Wei Shao, Qi Shao, Hai-Huan Xu, Guan-Ji Qiao, Run-Xuan Wang.

**Methodology:** Qi Shao, Hai-Huan Xu, Guan-Ji Qiao, Zhi-Yun Ma, Wei-Wei Meng.

**Project administration:** Wen-Wei Shao, Qi Shao, Xiaohong Li.

**Supervision:** Yun-Liang Zang, Xiaohong Li.

**Validation:** Hai-Huan Xu, Guan-Ji Qiao, Run-Xuan Wang, Zhuo-Bin Yang.

**Visualization:** Qi Shao, Run-Xuan Wang.

**Writing – original draft:** Wen-Wei Shao, Qi Shao.

**Writing – review & editing:** Wen-Wei Shao, Qi Shao, Hai-Huan Xu, Zhuo-Bin Yang, Yun-Liang Zang, Xiaohong Li.

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
