## [Decision Letter · Decision Letter 0]

9 Dec 2024

PCOMPBIOL-D-24-01467

Repetitive training enhances the pattern recognition capability of cultured neural networks

PLOS Computational Biology

Dear Dr. Li,

Thank you for submitting your manuscript to PLOS Computational Biology. After careful consideration, we feel that it has merit but does not fully meet PLOS Computational Biology's publication criteria as it currently stands. Therefore, we invite you to submit a revised version of the manuscript that addresses the points raised during the review process.

All three reviewers have indicated positive reactions to your findings, but also have pointed out some issues that should be clarified in the revision. Please be sure to address each of the reviewer comments in your resubmission. I draw special attention to reviewer 1's notes on the analysis of the spike signals, and point about control, and to reviewer 2's comments about mechanistic explanation and the analysis of the connectivity readout. 

Please submit your revised manuscript within 60 days Feb 08 2025 11:59PM. If you will need more time than this to complete your revisions, please reply to this message or contact the journal office at ploscompbiol@plos.org. Please include the following items when submitting your revised manuscript:

We look forward to receiving your revised manuscript.

Kind regards,

Alex Leonidas Doumas

Academic Editor

PLOS Computational Biology

Andrea E. Martin

Section Editor

PLOS Computational Biology

Feilim Mac Gabhann

Editor-in-Chief

PLOS Computational Biology

Jason Papin

Editor-in-Chief

PLOS Computational Biology

**Journal Requirements:**

At this stage, the following Authors/Authors require contributions: Wenwei Shao, Qi Shao, HaiHuan Xu, Zhiyun Ma, Guanji Qiao, Weiwei Meng, Zhuobin Yang, Runxuan Wang, Yunliang Zang, and Xiaohong Li. Please ensure that the full contributions of each author are acknowledged in the "Add/Edit/Remove Authors" section of our submission form.

Potential Copyright Issues:

- Please confirm that you are the photographer of Figure 1A, or provide written permission from the photographer to publish the photo(s) under our CC BY 4.0 license.

6) We note that your Data Availability Statement is currently as follows: "No data was used for the research described in the article.". Please confirm at this time whether or not your submission contains all raw data required to replicate the results of your study. Authors must share the “minimal data set” for their submission. PLOS defines the minimal data set to consist of the data required to replicate all study findings reported in the article, as well as related metadata and methods (https://journals.plos.org/plosone/s/data-availability#loc-minimal-data-set-definition).

- The points extracted from images for analysis..

7) Please amend your detailed Financial Disclosure statement. This is published with the article. It must therefore be completed in full sentences and contain the exact wording you wish to be published. Please ensure that the funders and grant numbers match between the Financial Disclosure field and the Funding Information tab in your submission form. Note that the funders must be provided in the same order in both places as well. State the initials, alongside each funding source, of each author to receive each grant. For example: "This work was supported by the National Institutes of Health (####### to AM; ###### to CJ) and the National Science Foundation (###### to AM)." State what role the funders took in the study. If the funders had no role in your study, please state: "The funders had no role in study design, data collection and analysis, decision to publish, or preparation of the manuscript.".

If you did not receive any funding for this study, please simply state: u201cThe authors received no specific for this work.u201d

**Reviewers' comments:**

Reviewer's Responses to Questions

**Comments to the Authors:**

Reviewer #1: This work studies the effect of training on culture networks and the association between evoked and spontaneous activity by looking at various properties of neural and network activity. The topic is interesting and experiments are meaningful

There are a few issues related to the presentation and design of the work. Hope the comments below are constructive.

1. One major point is that extracting spikes from the MEA data could have some problems. Using a threshold cutoff for spikes was not proper to separate different neurons. It is necessary to do spike sorting to have better the quality of neurons. Otherwise, they are more like MEA activity rather than single neurons. As such, functional network connectivity could have prudential issues. Since the data was collected, could be good to do spike sorting and get a better resolution of single neurons. Given the focus here is to study underlying neuronal mechanisms, using a better way of spike sorting is necessary.

2. When comparing the spontaneous activity of the network before and after training, could be better to have control cases, where there is no training applied but have the same time interval (say a few hours similar to the training time) of neural development. It is well known that neuronal tissue can develop quickly to generate essential network activity.

3. The abstract could be better revised. “cultured neural networks in vitro have demonstrated biological intelligence” sounds strange. Cultured tissues are biological. Intelligence was used in many locations but without proper definition. Classification patterns are not intelligence. The study here is to classify the neuronal pattern given different stimuli.

4. There are many places with overstated claims. Particularly, the text about AI is overstated, while the focus here is only related to playing cultured networks with various stimulations.

5. When using six stimulations, it may be necessary to have other cases for comparison For example, the same order and different orders, or random order of stimulation.

6. Citing references should have the same format: sometimes using 1,2 and 3, but sometimes using Yang et al.,2023 at line 67

7. Fig 1b: the authors state that both activities are similar at line 107. It is not very clear how they are similar. Maybe it is necessary to add a few characteristics for both signals.

8. Fig 1c is not clear: what is the green period for?

9. Fig.3: what is the different stimulation protocol used here?

Reviewer #2: Comments to the authors have been uploaded as attachment as a PDF file, named comments to the authors.

Reviewer #3: This is a study of how intelligence in pattern classification can arise from cultured neurons. This is a timely study that shows a new capability of intelligence to brain organoids.

I found the study to be wonderful and have slight suggestions, some just out of curiosity.

1. Another study that might be of interest in showing what brain organoids can do is this one: https://academic.oup.com/pnasnexus/article/2/6/pgad188/7202378

2. Figure 1f: I'm not sure what the x and y axes are. Maybe put in caption?

3. Before 2.2, maybe say you'll do more inputs? Supposedly a neural network has a memory of 1.4N/log N neurons or some such with the outer product rule, according to Hopfield's famous result-- we aren't coming close to that, but with that in mind, 2 seems somewhat unimpressive, but the results are actually impressive, so maybe just say that later on you do more patterns?

4. Line 150: Maybe try this method as well, or mention it? https://www.nature.com/articles/s41598-022-13674-4

5. Line 188: Maybe say that the parameter that's causing the tradeoff is the number of classes?

6. Figure 4-- just curious about how much better you can do with something more complicated than logistic regression.

7. Line 340: Cultured to cultured

8. Line 461: Can you talk about the pros and cons between your method and the method of Joost et al earlier?

**Have the authors made all data and (if applicable) computational code underlying the findings in their manuscript fully available?**

Reviewer #1: **No: ** no data were provided

Reviewer #2: **No: ** The authors declared the no data were used for the manuscript, but they talk about electrophysiological recording from cultured neurons. Data on these recordings, at this stage, are not made available.

Reviewer #3: Yes

PLOS authors have the option to publish the peer review history of their article (what does this mean? ). If published, this will include your full peer review and any attached files.

**Do you want your identity to be public for this peer review?** For information about this choice, including consent withdrawal, please see our Privacy Policy .

Reviewer #1: No

Reviewer #2: No

Reviewer #3: No

**Figure resubmission:**
---

## [Decision Letter · Decision Letter 1]

30 Mar 2025

PCOMPBIOL-D-24-01467R1

Repetitive training enhances the pattern recognition capability of cultured neural networks

PLOS Computational Biology

Dear Dr. Li,

Thank you for submitting your manuscript to PLOS Computational Biology. You will see that the reviewers are all in agreement the paper is much improved and makes a real contribution to the literature. Reviewer 2 raises a few minor points that should be addressed before publication. Please address these points and detail in a cover letter how they were addressed in the revised manuscript. It is unlikely the revision will have to go out for further review. 

Please submit your revised manuscript within 30 days May 30 2025 11:59PM. If you will need more time than this to complete your revisions, please reply to this message or contact the journal office at ploscompbiol@plos.org. Please include the following items when submitting your revised manuscript:

We look forward to receiving your revised manuscript.

Kind regards,

Alex Leonidas Doumas

Academic Editor

PLOS Computational Biology

Andrea E. Martin

Section Editor

PLOS Computational Biology

**Journal Requirements:**

1) Please amend your detailed Financial Disclosure statement. This is published with the article. It must therefore be completed in full sentences and contain the exact wording you wish to be published.

1) State the initials, alongside each funding source, of each author to receive each grant. For example: "This work was supported by the National Institutes of Health (####### to AM; ###### to CJ) and the National Science Foundation (###### to AM).".

**Reviewers' comments:**

Reviewer's Responses to Questions

Reviewer #1: The revision has included a substantial amount of new results, greatly enhancing the overall quality of the work. These additions provide deeper insights and strengthen the findings, making a significant contribution to the study.

Reviewer #2: I believe the manuscript has improved considerably, and all the previous issues have been addressed. However I have still some additional comments.

Major comments:

1- In the discussion, line 363, the author now makes a comparison with a previous work from Lamberti et al. ( reference number 35 in the manuscript). In this work Lamberti et al explain the use of a novelle technique to analyze functional connectivity in neuronal networks based on maximum entropy models (MaxEnt). This new method was compared to a more commonly used one that is conditional firing probability (CFP). In this work it came out that MaxEnt is a valuable alternative that can also distinguish between excitatory and inhibitory connections, something that CFP can’t do because it mainly relies on excitatory connections. The author instead writes “… Conditional Firing Probability (CFP) models to further characterize connectivity, particularly in detecting inhibitory connections…” I believe they were referring to the MaxEnt method and not the CFP one. If this is not the case can the author clarify how CFP detects inhibitory connections?

Minor comments:

1- In figure 1 line 136 the first sentence needs rephrasing. I believe that cultures were plated on MEAs at DIV 0 and used at DIV 21.

2- In the discussion line 399 and 404 the manuscript referenced ( number 41 in the reference list) has as first author Lamberti M. thus it should be referenced as Lamberti et al and not Joost et al.

Reviewer #3: All my critiques have been satisfied.

**Have the authors made all data and (if applicable) computational code underlying the findings in their manuscript fully available?**

Reviewer #1: None

Reviewer #2: Yes

Reviewer #3: Yes

PLOS authors have the option to publish the peer review history of their article (what does this mean? ). If published, this will include your full peer review and any attached files.

**Do you want your identity to be public for this peer review?** For information about this choice, including consent withdrawal, please see our Privacy Policy .

Reviewer #1: No

Reviewer #2: No

Reviewer #3: No

**Figure resubmission:**
---

## [Editor Report · Decision Letter 2]

10 Apr 2025

Dear Prof. Li,

We are pleased to inform you that your manuscript 'Repetitive training enhances the pattern recognition capability of cultured neural networks' has been provisionally accepted for publication in PLOS Computational Biology.

Best regards,

Alex Leonidas Doumas

Academic Editor

PLOS Computational Biology

Andrea E. Martin

Section Editor

PLOS Computational Biology

---

## [Editor Report · Acceptance letter]

PCOMPBIOL-D-24-01467R2

Repetitive training enhances the pattern recognition capability of cultured neural networks

Dear Dr Li,

I am pleased to inform you that your manuscript has been formally accepted for publication in PLOS Computational Biology. Your manuscript is now with our production department and you will be notified of the publication date in due course.

With kind regards,

Anita Estes
